# Meropenem PK/PD Variability and Renal Function: “We Go Together”

**DOI:** 10.3390/pharmaceutics15092238

**Published:** 2023-08-30

**Authors:** Jacopo Angelini, Simone Giuliano, Sarah Flammini, Alberto Pagotto, Francesco Lo Re, Carlo Tascini, Massimo Baraldo

**Affiliations:** 1Clinical Pharmacology and Toxicology Institute, University Hospital Friuli Centrale ASUFC, 33100 Udine, Italy; francesco.lore@asufc.sanita.fvg.it (F.L.R.); massimo.baraldo@uniud.it (M.B.); 2Department of Medicine, University of Udine (UNIUD), 33100 Udine, Italy; carlo.tascini@uniud.it; 3Infectious Diseases Division, Department of Medicine, University of Udine and Azienda Sanitaria Universitaria Friuli Centrale, 33100 Udine, Italy; simone.giuliano@asufc.sanita.fvg.it (S.G.); sarah.flammini@asufc.sanita.fvg.it (S.F.); alberto.pagotto@asufc.sanita.fvg.it (A.P.)

**Keywords:** therapeutic drug monitoring, meropenem, dose optimization, antibiotics, continuous infusion, renal function, pharmacokinetic, clinical pharmacology, critical illness, antimicrobial resistance

## Abstract

Background: Meropenem is a carbapenem antibiotic widely employed for serious bacterial infections. Therapeutic drug monitoring (TDM) is a strategy to optimize dosing, especially in critically ill patients. This study aims to show how TDM influences the management of meropenem in a real-life setting, not limited to intensive care units. Methods: From December 2021 to February 2022, we retrospectively analyzed 195 meropenem serum concentrations (Css). We characterized patients according to meropenem exposure, focusing on the renal function impact. Results: A total of 36% (*n* = 51) of the overall observed patients (*n* = 144) were in the therapeutic range (8–16 mg/L), whereas 64% (*n* = 93) required a meropenem dose modification (37 patients (26%) underexposed; 53 (38%) overexposed). We found a strong relationship between renal function and meropenem concentrations (correlation coefficient = −0.7; *p*-value < 0.001). We observed different dose-normalized meropenem exposure (Css/D) among renal-impaired (severe and moderate), normal, and hyperfiltrating patients, with a median (interquartile range) of 13.1 (10.9–20.2), 7.9 (6.1–9.5), 3.8 (2.6–6.0), and 2.4 (1.6–2.7), respectively (*p*-value < 0.001). Conclusions: Meropenem TDM in clinical practice allows modification of dosing in patients inadequately exposed to meropenem to maximize antibiotic efficacy and minimize the risk of antibiotic resistance, especially in renal alterations despite standard dose adaptations.

## 1. Introduction

Meropenem is one of the most widely used antibiotics, which is usually employed also as empirical and first-line anti-infective pharmacological therapy for the treatment of many kinds of infections in different clinical scenarios, including intensive care units. Due to the hydrophilic properties of beta-lactam antibiotics, of which meropenem is one, this drug is very prone to large pharmacokinetic (PK) and pharmacodynamic (PD) variability related to the most common alterations that occur during infections, especially severe ones, such as increased volume of distribution, cardiac output, capillary leakage, impaired or augmented renal function, hypoalbuminemia, and any other pathophysiological modification induced by a more or less extensive and lasting inflammatory response [1]. All these conditions could significantly jeopardize the appropriate exposure to meropenem, with a high risk for patients to be exposed to unstable serum concentrations of the drug throughout the duration of antibiotic therapy. Meropenem is characterized by a time-dependent antimicrobial effect, like all other beta-lactam agents, and therefore it is necessary for its plasma concentrations to durably remain above the minimal inhibitory concentration (MIC) for as long as possible during the dosing interval (100% T > MIC) to increase the probability of the success of the therapy [2]. To achieve this, one of the most helpful therapeutic strategies is represented by the continuous infusion of the antibiotic [3,4]. On the other hand, in case of any concomitant and intervening condition that can determine a reduction in antibiotic concentration, especially in the first phases of infection, this could lower the probability of survival or clinical cure [5]. Usually, since meropenem is mainly eliminated by renal excretion, serum creatinine and estimated renal clearance are two of the most monitored parameters by physicians who have to face the risk of under- or over-exposure of their patients. Therefore, the adequate adjustment of the doses of antibiotics, meropenem included, still today represents a critical matter in the management of drug therapy, so much so that mortality and increasing antimicrobial resistance remain two important unmet medical needs in clinical practice [6,7,8,9]. Therefore, an increasing amount of scientific evidence in the literature recommends performing the therapeutic drug monitoring (TDM) of beta-lactams, especially in critical care and in case of alterations of dysfunction of organs involved in the absorption, distribution, metabolism, and excretion of administered drugs [9,10]. Nevertheless, the TDM and the clinical pharmacologist advice is available only in a limited number of hospitals, mainly due to several organizational barriers to the implementation of this routine activity in terms of costs and dedicated staff [11].

A routine service for the TDM of meropenem, and other antimicrobics, was performed at our hospital by the Clinical Pharmacology Institute of the “Santa Maria della Misericordia Hospital”, of Udine, Italy. We retrospectively investigated the meropenem TDM performed by our institute both to analyze our activity for the dose optimization of antimicrobics and to further characterize the pharmacokinetic properties of meropenem by evidence emerging from a real-world setting.

## 2. Materials and Methods

This was a monocentric, retrospective study investigating the effectiveness of different dosage regimens of meropenem administered by continuous infusion in routine clinical practice combined with the TDM service performed by the Clinical Pharmacological Institute in our hospital.

Demographic, clinical, and laboratory features were documented between December 2021 and February 2022. Due to the retrospective and observational nature of the analysis, written consent was waived according to our institution’s agreements.

Patients aged 18 years or above, admitted to all units of the “*Santa Maria della Misericordia Hospital*” in Udine, and treated with meropenem—administered by continuous infusion preceded by a loading dose—and TDM required according to clinical practice were included. The employment of meropenem was at the discretion of the treating physician, who prescribed the starting dose according to local guidelines and clinical conditions. We considered only patients who achieved meropenem serum concentrations at the steady state, including patients who received a meropenem bolus before the administration of continuous infusion according to local guidelines and/or who started the meropenem therapy at least 24 h before the meropenem TDM sampling, based on medical records. Meropenem serum concentrations were excluded when steady-state concentrations were not achieved. Patients without main clinical, laboratory, or pharmacological parameters such as BMI, renal function, or meropenem doses or concentrations were excluded from the correlation and comparison statistical tests. Patients who underwent additional TDM of meropenem subsequent to the initial assessment for the same clinical condition were included in the study. The two meropenem concentrations at the steady state were utilized to estimate the relevant PK/PD target achievements. A comparison between these two concentrations was conducted using a Sankey plot. Since meropenem was administered as a continuous infusion, the exposure to meropenem was estimated taking into consideration the meropenem serum concentration at the steady state (Css) and calculating the daily area under the plasma drug concentration–time curve (eAUC_0–24h_) as 24-fold the Css. Consistently, meropenem clearance (meropenem CL) was estimated as the ratio between the administered daily dose and the pertinent eAUC_0–24h_.

According to the summary of the product characteristics (SmPC) of meropenem, the creatinine clearance (CLCr) was estimated by the Cockroft–Gault equation. Patients were stratified based on estimated creatinine clearance as follows: group 1: <10 mL/min; group 2: 10–25 mL/min; group 3: 26–50 mL/min; group 4: 50–120 mL/min; and group 5: >120 mL/min. Groups 1–3 represented patients with renal impairment according to the classification that the SmPC of meropenem suggests to consider for antibiotic dose adjustment, whereas groups 4 and 5 represented patients with normal and augmented glomerular function, respectively.

Meropenem blood samples were immediately transferred to the laboratory, processed according to local guidelines of the Clinical Pharmacology Institute, and measured using the High-Performance Liquid Chromatography-Ultraviolet (HPLC-UV) method. We analyzed serum meropenem concentrations using a validated method, with some modifications we described previously [12,13,14]. Precision and accuracy were evaluated by replicate analyses of quality control samples against calibration standards. Intra- and inter-assay coefficients of variation were always <10%. The lower limit of detection was 0.5 mg/L.

The other laboratory parameters were determined by standard clinical chemical methods. 

We chose the MIC used for PK/PD target attainment determination based on one of the most common worst-case scenarios: 2 mg/L, corresponding to the ECOFF of *Pseudomonas aeruginosa* according to the European Committee on Antimicrobial Susceptibility Testing (EUCAST) [15,16]. A threshold of potential meropenem toxicity was set at 44.5 mg/L according to the literature.

Statistical analyses were performed with Prism statistics (GraphPad Software version 10.0, San Diego, CA, USA). The Shapiro–Wilk normality test was performed for each data set. Continuous variables were described using median and interquartile range (IQR), and categorical variables were described using counts and frequencies. *p*-values for the comparison of different subgroups were derived from the Kruskal–Wallis test for continuous variables with a post-hoc Dunn’s multiple comparison test. Agreement between X and Y was assessed by Spearman’s rank correlation because of the not-normal distribution of analyzed variables. The Spearman’s coefficient of correlation (r) was considered relevant when > 0.5. Contingency analysis was performed by Fisher’s exact test and the result was presented as odds ratio (OR). Correlation among categorical variables was determined by the chi-square test. Statistical significance was set for a two-tailed *p*-value ≤ 0.5.

## 3. Results

### 3.1. Demographic and Clinical Data

The detailed description of the main baseline demographic and clinical features of the 144 patients included in the analyses are reported in Table 1. Out of the total number of patients, 35.4% (*n* = 51) performed a second meropenem TDM after a median of 96 h (IQR: 72–168) from the first TDM, resulting in a total of 195 meropenem serum concentrations measured.

The patients were hospitalized in the following wards: intensive care (*n* = 36; 25%), internal medicine (*n* = 36; 25%), hematology (*n* = 14; 10%), infectious disease (*n* = 11; 8%), general surgery (*n* = 8; 6%), hepatology (*n* = 8; 6%), cardiovascular surgery (*n* = 6; 4%), neurosurgery (*n* = 6; 4%), orthopedic (*n* = 5; 3%), other wards for the remaining patients (*n* = 11; 8%), and for 3 patients (1%), this kind of information was not available. The most frequently diagnosed pathologies were (in descending order) complicated intra-abdominal infection, febrile neutropenia, and central nervous system infection. A total of 17 patients (11.8%) were diagnosed with sepsis and 3 patients (2.1%) with septic shock. On the first day of meropenem TDM, no patients were with renal replacement therapy.

### 3.2. Patients’ Exposure to Meropenem

Meropenem was administered as a continuous infusion in all patients with a maintenance daily dose of 4 g in 39 patients (27.1%), 2 g in 39 patients (27.1%), 3 g in 31 patients (21.5%), 6 g in 16 patients (11.1%), 1 gr in 16 patients (11.1%), 1.5 g in 4 patients (2.8%), 0.5 g in 3 patients (2.1%), and 8 g in 1 patient (0.7%). The median total meropenem serum concentration at the steady state was 13.56 mg/L (IQR: 7.8–19.1) (Figure 1a). The median exposure to meropenem represented by the serum antibiotic concentration at the steady state (Css), which was normalized by the administered daily dose (D) (Css/D ratio), was 4.7 (IQR: 2.5–8). The median estimated daily exposure to meropenem (estimated area under the plasma drug concentration–time curve; eAUC_0–24h_) and meropenem clearance (CL) were 325.44 mg × h/L (IQR:187.2–457.9) and 146.89 mL/min (IQR: 87.1–275.1), respectively. The pharmacological target of 100% > T_4–8xMIC_ in the first days of the infection was adequately attained by 51 patients (36%), whereas the remainder of them (64%) were overexposed or underexposed (Figure 1b).

Patients who were underexposed to meropenem (<8 mg/L) were younger than overexposed patients (*p*-value = 0.002), and they had lower serum creatinine than patients overexposed to meropenem (*p*-value < 0.001). Furthermore, overexposed patients had higher C-reactive protein levels (*p*-value = 0.004) and higher serum creatinine than patients with meropenem concentration within the therapeutic range (*p*-value = 0.002), and they consistently showed lower estimated glomerular filtration compared to patients with adequate meropenem concentration (*p*-value = 0.001) and underexposed patients (*p*-value < 0.001). Similar median daily doses of meropenem were infused among patients attaining and not attaining the PK/PD meropenem target, despite statistically significant differences in terms of exposure among patients on target, under-, and overexposed, as summarized by the ratio between the measured serum concentrations at the steady state of meropenem and the daily infused dose (Css/D). These differences were confirmed when the meropenem concentrations were normalized by body weight (C/kg) and both daily dose and body weight (C/D/kg) (Table 2).

### 3.3. Meropenem Exposure and Renal Function

Due to the relevant glomerular excretion of meropenem, a strong correlation between the dose-normalized steady-state serum concentration of meropenem and serum creatinine (Figure 2a) and between the drug clearance and creatinine clearance was found (Figure 2b).

Consistently, when we took into consideration the groups of patients stratified by estimated glomerular function, a progressively higher Css/D ratio can be appreciated for progressively lower estimated creatinine clearance (Figure 3a). Median and interquartile ranges of Css/D decreased from group 2 to group 5: 13.1 (10.9–20.2), 7.9 (6.1–9.5), 3.8 (2.6–6.0), and 2.4 (1.6–2.7), respectively. Statistically significant differences are reported for all the groups (*p*-value < 0.001), except for patients in group 2 vs. group 3. Statistically significant mean rank differences are the following: group 2 vs. group 4 and 5: 61.1 and 90.1, respectively; group 3 vs. group 4 and 5: 30.1 and 59.2, respectively; group 4 vs. group 5: 29.1. Similar results can be observed when meropenem clearance is investigated (Figure 3b), fostering the strong reverse relationship between glomerular function and meropenem exposure. In this case, we observed increasing meropenem clearance across the groups, with the following median and interquartile ranges from group 2 to group 5: 53.0 mL/min (34.4–63.8), 80.6 mL/min (70.2–100.3), 172.8 mL/min (114.3–261.1), and 276.7 mL/min (232.5–365.5), respectively. Similar statistical evidence that was found from Css/D analysis was also confirmed for meropenem clearance (*p*-value < 0.001), with the following statistically significant mean rank differences: group 2 vs. group 4 and 5 = −60.6 and −81.8, respectively; group 3 vs. group 4 and 5 = −42.5 and −63.7, respectively; and group 4 vs. group 5 = −21.2. No statistically significant differences were found between group 2 vs. group 3 and between group 4 and group 5. Analog results can be described when the serum meropenem concentrations are normalized for weight (*p*-value < 0.001) and for both daily dose and weight (*p*-value < 0.001).

The different exposure to meropenem consequently also influenced the PK/PD target attainment. This was shown when the median of antibiotic concentration at the steady state was compared among the groups. Higher meropenem concentrations were observed in patients with lower creatinine clearance, though they received significantly reduced doses compared to patients with higher glomerular filtrations (Figure 4a,b). Statistically significant differences in meropenem concentrations were observed for the following groups: (a) patients with CLCr between 10 and 25 mL/min (median: 15.7 mg/L; IQR: 12.7–23.9) compared to patients with >120 mL/min (median: 9.0 mg/L; IQR: 6.9–11.6) (*p*-value = 0.002), with a statistically significant mean rank difference of 46.9; (b) patients with CLCr between 26 and 50 mL/min (median: 18.7 mg/L; 13.1–24.7) compared to both patients with CLCr between 50 and 120 mL/min (median: 13.8 mg/L; IQR: 7.6–18.4) (*p*-value = 0.047) and patients with >120 mL/min (*p*-value < 0.001), with a statistically significant mean rank difference of 23.2 and 48.1, respectively; and (c) patients with CLCr between 50 and 120 mL/min compared to patients with >120 mL/min (*p*-value = 0.023), with a statistically significant mean rank difference of 24.9. Patients with CLCr ranging from 10 to 25 mL/min received a lower meropenem daily dose (median 1 gr; IQR: 1–2) compared with any other patients (*p*-value < 0.001) (median patient with normal renal function: 3 gr; IQR: 2–4; median hyperfiltrating patients: 4 gr; IQR: 3–6; with a statistically significant mean rank difference of −117.2 and −154.6, respectively), except for patients with moderate renal impairment (median 2 gr; IQR: 2–3.25). No differences were demonstrated in terms of both administered daily dose and meropenem exposure among patients with renal impairment ranging from 10 mL/min up to 50 mL/min (Figure 4a,b).

We investigated the number of patients who achieved the PK/PD target in each renal function group. The prevalence of patients achieving PK/PD target in the different groups was 46% (*n* = 6) in group 2, 23% (*n* = 7) in group 3, 34% (*n* = 21) in group 4, and 45% (*n* = 15) in group 5. The prevalence of patients underexposed to meropenem was 8% in group 2 (*n* = 1), 13% in group 3 (*n* = 4), 28% in group 4 (*n* = 17), and 42% (*n* = 14) in group 5. The prevalence of patients overexposed to meropenem was 46% (*n* = 6) in group 2, 63% (*n* = 19) in group 3, 38% (*n* = 23) in group 4, and 12% in group 5. Consistently, most patients with renal impairment (CLCr < 50 mL/min) were characterized by an overexposure to meropenem compared to patients with normal or augmented renal function, despite the adjustment of the antibiotic dose according to their renal function. Moreover, patients with high glomerular filtration activity (>120 mL/min) were characterized by a low prevalence of overexposure to meropenem compared to patients with normal or impaired renal function, although they received increased meropenem doses (chi-square test = 18.53, *p*-value < 0.001) (Figure 5a). Overall, despite a dose adjustment of meropenem according to changes in renal function, at the first TDM measurement since the start of the setting of anti-infective therapy, patients with both increased and decreased glomerular filtrate changes had increased risk of failing to achieve adequate drug concentrations in the blood, compared with patients with renal function between 50 and 120 mL/min (odds ratio: 5.6, 95% CI 2.4–12.0, *p*-value < 0.0001) (Figure 5b).

### 3.4. Meropenem Exposure and Other Clinical Data (Sex, BMI, and Age)

Taking into consideration sex, no differences were observed in terms of the following: (a) meropenem serum concentrations at the steady state, with a median (IRQ) of 12.72 mg/L (7.74–18.45) in male vs. 14.70 mg/L (7.80–20.59) in female patients (*p*-value = 0.18); (b) daily dose of meropenem infused with a median (IQR) of 3 gr (2–4) (*p*-value = 0.28); (c) dose-normalized exposure to meropenem with a median (IQR) of 3.9 (2.5–8.1) in male and 5.5 (2.5–8.0) in female patients (*p*-value = 0.21); and (d) meropenem clearance with a median (IQR) of 176.5 mL/min (85.6–281.8) in male and 125.1 mL/min (87.1–275.0) in female patients (*p*-value = 0.21). No correlations were found between BMI and meropenem serum concentration, Css/D, or meropenem clearance.

Focusing on age, we stratified patients into three groups: under 50 years old, between 50 and 75 years old, and over 75 years old. We found significantly lower creatinine clearance values in the elderly patient groups (*p*-value < 0.001). The median (IQR) estimated CLCr was 164.2 mL/min (114–212.7), 83.0 mL/min (52.7–122.5), and 45.9 mL/min (28.6–77.3) in the youngest group, middle group, and oldest group, respectively. We observed a significant difference in terms of exposure to meropenem (*p*-value = 0.009), since patients older than 75 years were overexposed to meropenem compared with younger patients with a median (IQR) of 16.3 mg/L (10.2–24.3), whereas patients under 50 years old and between 50 and 75 years old showed median (IQR) meropenem concentrations of 9.6 mg/L (7.2–14.5) and 12.4 mg/L (7.7–18.2), respectively. The same statistically significant differences (*p*-value < 0.001) were observed also for the Css/D (median 7.4 (4.1–11.5) vs. 3.7 (2.5–6.8) in patients between 50 and 75 years old and 2.4 (1.8–3.2) in patients < 50 years old) and for the meropenem clearance (median 94.3 (60.2–168.5) vs. 189.0 (102.3–281.1) in patients between 50 and 75 years old and 292.1 (221.0–380.0) in patients < 50 years old). 

### 3.5. The Support of the Therapeutic Drug Monitoring in the Meropenem Dosing Management

Of the initial 144 patients for whom an initial TDM was requested, further measurement of meropenem serum concentrations was required for 51 of them (35.4%): 34 male vs. 17 female (66.7% vs. 33.3%), with a median age (IQR) of 72 (59–78) years old, a median BMI (IQR) of 24.2 (23.1–27.7) kg/m^2^, and most of them were admitted in intensive care units (n = 14; 27.4%) and internal medicine wards (n = 12; 23.5%).

Based on the interpretation of the meropenem serum concentration and by the analysis of the clinical and laboratory conditions of each patient, The Clinical Pharmacology Institute provided advice to the treating physician regarding antibiotic dose adjustment. In 21 patients (41.2%), no dose adjustments were indicated, whereas in 19 patients (37.2%) and 11 patients (21.6%), a dose reduction and a dose increase were suggested, respectively. In 29 patients out of 51 (56.9%), the treating physician followed the advice provided after TDM. Overall, the meropenem daily dose was modified in 19 patients (37.2%) as follows: a dose reduction was performed in 12 patients (63.2%) (median reduction of 1 gr), whereas in 7 patients (36.8%), the daily dosage of meropenem was increased (median of 2 gr). A second TDM was performed after a median of 96 h (IQR: 72–168) from the first TDM. Overall, after the support of meropenem TDM, the prevalence of patients underexposed and overexposed to the antibiotic decreased, whereas the number of patients who achieved the therapeutic concentrations increased (Figure 6).

## 4. Discussion

To the best of our knowledge, this is one of the largest retrospective analyses of the employment of TDM to optimize antimicrobial therapy in our hospital. Unlike other similar studies, this analysis adds new evidence regarding the exposure to meropenem in patients admitted to non-intensive care unit wards, suggesting that inadequate antibiotic doses could also be administered in this scenario.

The use of TDM is recommended by several scientific societies and healthcare professionals, who suggest performing daily TDM of beta-lactams in intensive care patients [9,17,18,19]. Nevertheless, relevant pathophysiological alterations that could influence the distribution and elimination of beta-lactams also happen in non-intensive care unit wards, and inadequate dose administration of antibiotics can occur. In our analyses, during the observation period, only 25% of meropenem TDMs were required by the intensive care unit, whereas the remaining part of patients were admitted to medical or surgical units. This means that in a real-life scenario, several clinical and therapeutic conditions such as age, kidney function, and variations of the volume of distribution mainly related to altered fluid balance could commonly influence some crucial pharmacokinetic determinants. These suppositions can be supported by the high prevalence of investigated patients (64%) who had meropenem serum concentrations outside the therapeutic range. This has two main consequences. First, TDM was able to promptly identify underexposed or overexposed patients, supporting treating physicians in meropenem dose adjustments to achieve a more appropriate PK/PD target. Secondly, the employment of meropenem TDM in clinical practice gave us the opportunity to set up an MIC-driven antimicrobial therapy. Since the use of carbapenems is the most significant risk factor associated with intestinal colonization by carbapenemase-producing *Enterobacteriales* (CPE) and the development of subsequent CPE bacteremia [20], it is crucial to use them judiciously, ensuring adequate patient exposure. TDM allows for adequate patient exposure in terms of killing and suppression of resistance development [1,21], thus avoiding potentially risky therapy for the development of gut CPE colonization that is also ineffective, adding insult to injury. Our results showed that only 2 patients presented potentially toxic meropenem serum concentrations, suggesting that there is a lower risk of safety issues compared to the higher risk of not administering adequate antibiotic doses (2% vs. 26% of patients), threatening the control of the infectious disease. However, from a clinical and pharmacological point of view, in the case of administration of labeled meropenem doses, most patients showed higher meropenem concentrations than required according to the EUCAST breakpoints. This suggests that more personalized antibiotic therapy can be pursued by a TDM-driven approach, which can dynamically adapt the therapy throughout the development of the infection. In this case, a possible positive consequence could be a reduction in administered meropenem doses, aiming both to obtain effective antibiotic concentrations and to lower the risk of antibiotic resistance induction. A valuable effect in terms of pharmacoeconomics could be supposed for the TDM approach, as suggested by some studies [22,23,24,25], although dedicated and specific investigations should be performed. Another significant finding from our data is that administering meropenem through continuous infusion, preceded by a loading dose, achieves therapeutic concentrations in most patients within a few hours of commencing meropenem therapy. This approach ensures continuous exposure to drug concentrations that exceed the MIC and effectively combats the pathogens from the early stages of infection, which is crucial for improving the clinical outcome of the patient [26,27,28,29]. In addition, the administration of meropenem throughout the dosing interval represents a good therapeutic strategy in order to maximize the time-dependent antimicrobial activity of beta-lactams, fostering the maintenance of drug serum concentration above the MIC (T > MIC). In this analysis, we chose a reference PK/PD target represented by meropenem concentration exceeding 4-fold the MIC. This target is a subject of debate, due to two main issues: (a) few prospective randomized clinical trials have been conducted to properly identify a robust threshold; (b) most of the evidence derives from in vitro or in vivo studies and retrospective observational trials, suggesting ranges from 1-fold the MIC to 8-fold the MIC [17,30,31]. We decided to choose a wide therapeutic range due to the heterogeneous clinical conditions of the observed patients, coming from different hospital wards. Nevertheless, we considered a more aggressive value of the PK/PD target (100% T > _4–8xMIC_), since one out of four of the observed patients were admitted to a critical care unit, where maximal bacterial killing activity is required, in order to increase the possibility of obtaining a positive clinical outcome and to avoid bacterial regrowth [22].

Focusing on the observed patients who did not achieve the PK/PD therapeutic target, our study confirms the few data in the literature about the relationship between renal function and meropenem exposure [32,33,34]. Patients with higher glomerular filtrations are characterized by lower meropenem exposure (correlation coefficient *r* = −0.7; *p*-value < 0.001). This relationship with glomerular filtration was also confirmed when we considered the meropenem serum concentrations normalized for the administered daily dose. We performed this analysis to exclude the possibility that the variations of meropenem concentrations were influenced by the dose of antibiotic taken by the patients. Since meropenem was administered by continuous infusion, we could directly estimate the drug clearance and we found a strong concordance with glomerular filtration (correlation coefficient *r* = −0.7; *p*-value < 0.001). This confirms the clinical and pharmacological importance of renal function, which is one of the most important parameters that significantly influences exposure to meropenem, and our data suggest physicians take into consideration this parameter, especially in hospitals where TDM is not available. In our study, we used the Cockroft–Gault equation to estimate creatinine clearance, which we chose as the surrogate outcome of glomerular filtration rate, although the measured creatinine clearance should be preferred in critically ill patients because it is more accurate [35]. The relevance of the impact of renal function is shown by the differences we evidenced in terms of meropenem concentrations, normalized for the daily dose administered, when we compared patients grouped by glomerular function. Patients characterized by moderate to severe renal impairment showed higher meropenem concentrations with lower drug clearance, although they received lower doses than other patients. On the other hand, a piece of more clinically relevant information derived from our analysis is that hyperfiltrating patients received the highest daily dose of meropenem compared to the other two groups, but the meropenem serum concentrations were the lowest and most of the measured antibiotic concentrations were near the lower limit of the PK/PD target. This should be taken into consideration by physicians, since in the case of patients with an estimated creatinine clearance > 120 mL/min, a condition of underexposure to meropenem could be more likely than in the case of normal or impaired renal function, despite a higher daily dose administered. This condition poses an insidious challenge in antimicrobial therapy, given the associated risks of negative clinical outcomes and antimicrobial resistance occurrence due to inadequate drug exposure. For this reason, the employment of TDM of meropenem could represent useful support in its dose management, as suggested by the comparison we performed on the drug concentrations between the first execution of TDM and the second one. We described a reduction in both underexposed and overexposed patients with a concomitant increase in patients on target. 

As mentioned before, meropenem is a very common first-line antimicrobial therapy method, especially in the case of empirical treatment in critically ill patients. Nevertheless, similar clinical conditions can require the use of other beta-lactam antibiotics, such as new beta-lactams/beta-lactamase inhibitors (BLBLI). Because of a relevant glomerular excretion of both meropenem and BLBLI, evidence reported in this analysis might likely be applied to these alternative antimicrobial combinations.

However, a very limited number of studies have been conducted to properly analyze the impact of TDM activity, due to the low prevalence of hospitals with a routine TDM clinical service and the complexity of identifying an adequate performance outcome [11,36]. Nevertheless, the possibility of measuring drug concentration throughout the dosing interval helps physicians modify the labeled dose regimen in order to exploit the PK/PD properties of the active principle, promptly monitoring possible unexpected therapeutic underexposure or overexposure of patients, meanwhile evaluating the clinical evolution of the disease. Recently, our group successfully adopted this approach for the treatment of endocarditis by *E. faecalis* with an extension of the labeled duration of the infusion of ceftobiprole, shifting from the two hours indicated in the summary of the product characteristics to three hours to maximize the time-dependent bactericidal properties of ceftobiprole [37].

Focusing on the other features of patients that we investigated, we did not find any specific differences in terms of exposure to meropenem, or target attainment based on sex, whereas patients grouped by age suggest a higher exposure to meropenem and target attainment in older patients compared to younger ones. Nevertheless, these data seem to be influenced by the differences in renal function that characterize the groups, and which are aligned with the relationship between glomerular filtration and meropenem exposure we previously discussed. Because of the retrospective nature of our analysis and the limited number of patients included in our study, we cannot speculate on the possible association between ethnicity and meropenem serum concentrations. Nevertheless, this aspect should be further investigated because evidence of the influence of ethnicity on renal function has been reported [38,39]. No significant impact of body weight or BMI was shown; however, our patients were not characterized by elevated weight and BMI, so no conclusions can be formulated from our data on overweight and obese patients, as instead suggested by Pai et al., who proposed a specific dose correction in these special populations [40].

This study has several limitations: (a) the retrospective nature of the study could negatively impact the accuracy of data selection and collection; (b) since this analysis was performed in a single center, a limited number of patients could be included; and (c) no clinical outcomes were investigated and no robust association with meropenem serum concentrations could be assessed.

Prospective trials focusing on the relationship between antibiotic exposure and clinical and laboratory features of patients are required. 

## 5. Conclusions

Meropenem is widely regarded as one of the most frequently used antibiotics in various clinical settings. The variability of meropenem is mainly linked to renal function, which represents one of the most common determining factors that can significantly influence the antibiotic exposure of patients. Despite the administration of an adjusted dose suggested from phase III trials, exposure to meropenem shows a relevant variability, not only in the critical care unit, but also in any other wards, since renal function alterations are very common in hospitalized patients. Therefore, the TDM of meropenem, such as of any other antibiotic, is a useful tool to support all physicians in the individuation of a personalized dose for all patients. The antibiotic treatment based on TDM could play a key role in the multidisciplinary action plans against severe or difficult-to-treat infections and antimicrobial resistance.

## Figures and Tables

**Figure 1 pharmaceutics-15-02238-f001:**
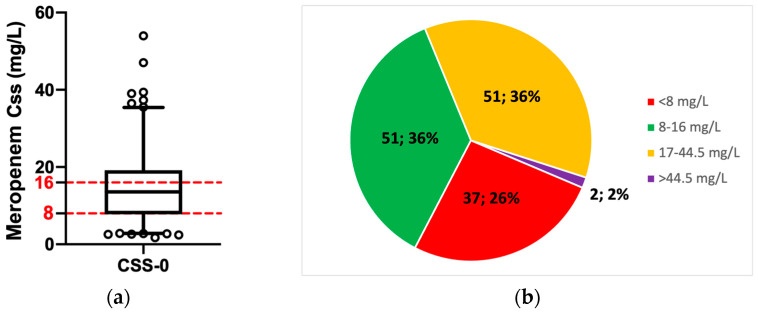
Patients’ exposure to meropenem: (**a**) Meropenem serum concentration and PK/PD target attainment at the first therapeutic drug monitoring required for patients treated with meropenem. Red dotted lines represent the therapeutic range for meropenem. (**b**) Prevalence of patients who achieved the PK/PD target and who did not. Target classification according to meropenem serum concentration: <8 mg/L = underexposed patients; 8–16 mg/L = at target patients; 17–44.5 mg/L = overexposed patients; >44.5 mg/L = patients exposed to meropenem toxic concentration. Css = meropenem serum concentrations at the steady state. Data are reported as median and percentile. Circles represent outliers.

**Figure 2 pharmaceutics-15-02238-f002:**
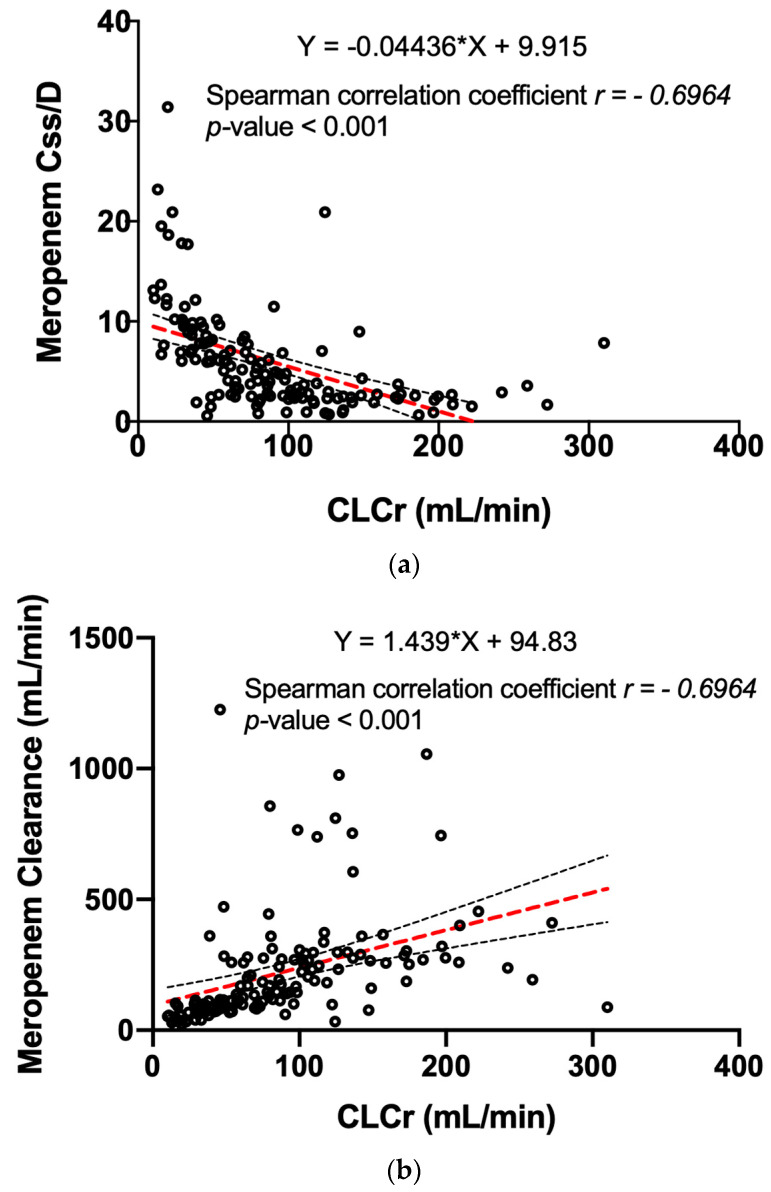
The correlation between creatinine clearance and meropenem exposure: (**a**) The distribution of meropenem steady-state concentrations normalized for the administered dose (Css/D) in relation to the serum creatinine clearance. The dotted lines represent the linear regression equation (in red) and the 95% CI (in black). (**b**) The direct correlation between the CLCr and the meropenem clearance. The dotted lines represent the linear regression equation (in red) and the 95% CI (in black). CLCr = creatinine clearance.

**Figure 3 pharmaceutics-15-02238-f003:**
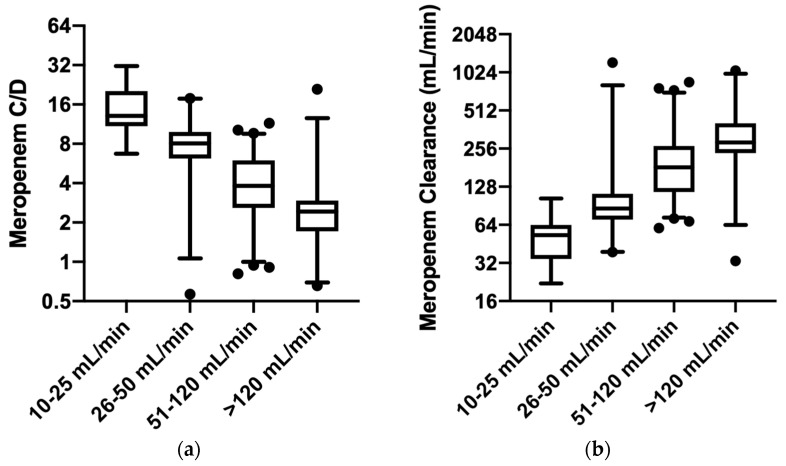
The meropenem exposure in the groups of patients stratified by different glomerular filtration: (**a**) The distribution of meropenem steady-state concentrations normalized for the administered dose (C/D) among patients with different renal functions estimated by creatinine clearance. (**b**) The meropenem clearance among patients with different renal functions estimated by creatinine clearance. The *y*-axis is represented on a log2 scale both in (**a**) and in (**b**). Median and interquartile range are presented. Black circles represent outliers.

**Figure 4 pharmaceutics-15-02238-f004:**
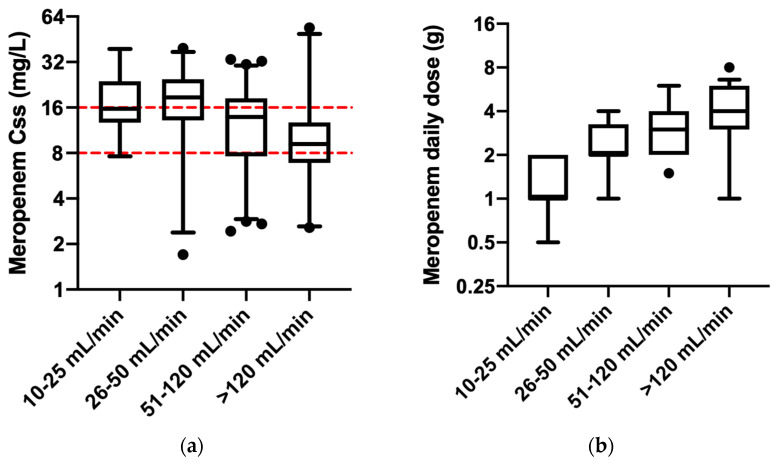
The PK/PD meropenem target attainment among the groups of patients stratified by different glomerular filtration and the pertinent meropenem daily dose: (**a**) The distribution of meropenem steady-state concentrations at the first therapeutic drug monitoring required for patients treated with meropenem. Red dotted lines represent the therapeutic range for meropenem. (**b**) The meropenem daily dose administered as a continuous infusion among patients with different renal functions. The y-axis is represented on a log2 scale both in (**a**) and in (**b**). Median and interquartile range are presented. Black circles represent outliers.

**Figure 5 pharmaceutics-15-02238-f005:**
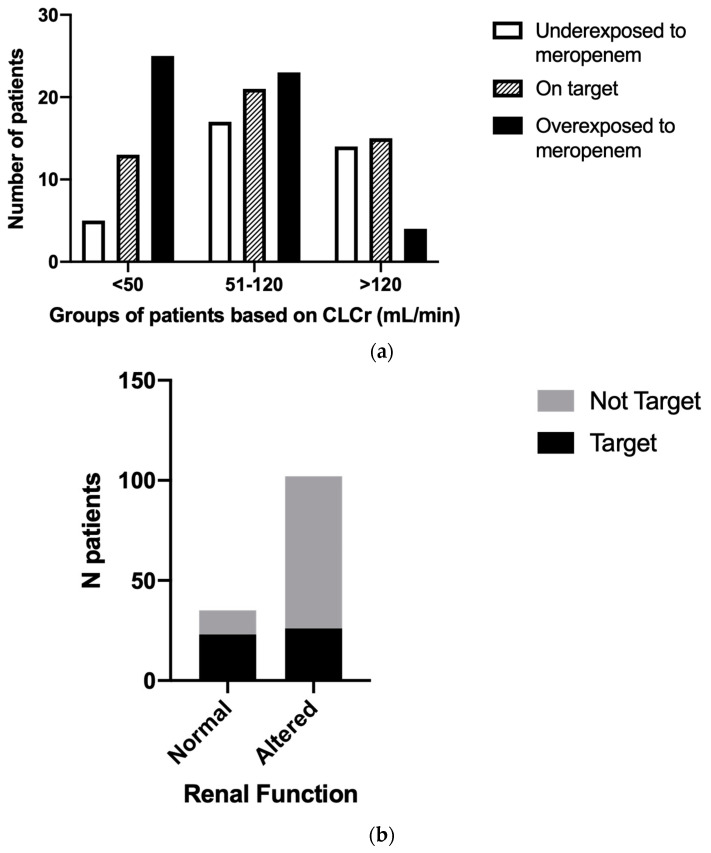
The prevalence of patients achieving the PK/PD meropenem target among different glomerular filtration classes. (**a**) Number of patients reporting serum meropenem concentrations under the therapeutic target concentrations (white columns), within the therapeutic range (lined columns), and over the therapeutic target concentrations (black columns). Patients were grouped according to their renal function: impaired (CLCr < 50 mL/min), normal (CLCr 50–120 mL/min), and augmented (>120 mL/min) renal function. (**b**) Number of patients reporting serum meropenem therapeutic concentrations (black bar) and not reporting serum meropenem therapeutic concentrations (gray bar). All patients were grouped in normal function in case of CLCr between 50 mL/min and 120 mL/min and altered in case of CLCr < 50 mL/min or > 120 mL/min. CLCr = creatinine clearance.

**Figure 6 pharmaceutics-15-02238-f006:**
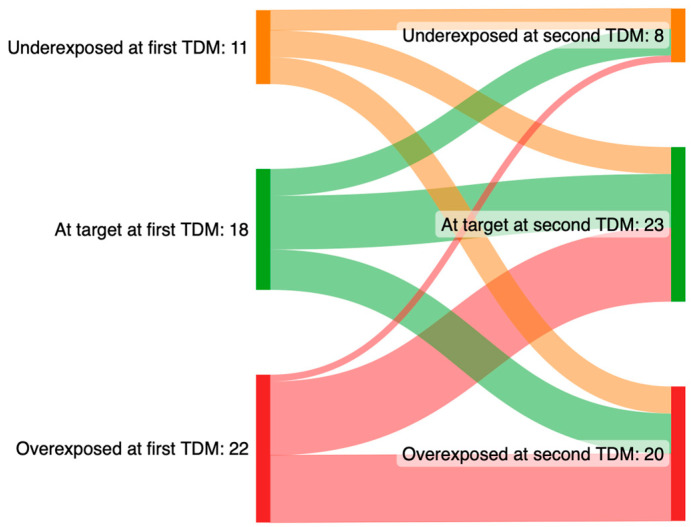
Sankey plots of pharmacological target attainment of meropenem between the first and the second therapeutic drug monitoring. Meropenem serum target concentrations: 8–16 mg/L; underexposure: <8 mg/L; overexposure: >16 mg/L.

**Table 1 pharmaceutics-15-02238-t001:** Demographic and clinical data at the first meropenem therapeutic drug monitoring.

Variable	n (%) or Median (IQR)
Sex (male/female)	92/52 (63.9/36.1)
Age (years)	72 (60–79)
Weight (kg)	75 (65–85)
Height (cm)	170 (164–178)
BMI (kg/m^2^)	25 (23–28)
Serum total protein (g/L)	56.0 (51.0–61.2)
Serum albumin (g/L)	28.5 (24.0–32.4)
Alanine aminotransferase (UI)	23.0 (13.2–42.5)
Serum creatinine (mg/dL)	0.90 (0.63–1.42)
Patients with eCLCr ^1^ < 10 mL/min	0 (0)
eCLCr ^1^ in < 10 mL/min group	-
Patients with eCLCr ^1^ 10–25 mL/min	14 (10.1)
eCLCr ^1^ in 10–25 mL/min group	16.25 (14.15–19.43)
Patients with eCLCr ^1^ 26–50 mL/min	30 (21.6)
eCLCr ^1^ in 26–50 mL/min group	30.15 (33.05–45.75)
Patients with eCLCr ^1^ 50–120 mL/min	62 (44.6)
eCLCr ^1^ in 50–120 mL/min group	81.05 (65.73–98.63)
Patients with eCLCr ^1^ >120 mL/min	33 (23.7)
eCLCr ^1^ in >120 mL/min group	159.10 (136.60–197.30)
C-reactive protein (mg/L)	58.3 (22.1–123.7)
Time between initiation therapy with meropenem and first TDM (hours)	96 (72–120)

^1^ eCLCr: estimated clearance of creatinine according to the Cockroft–Gault formula.

**Table 2 pharmaceutics-15-02238-t002:** Patient characteristics grouped by attainment status of the pharmacological target meropenem serum concentrations at the first meropenem therapeutic drug monitoring.

Variable ^1^	Patients under the PK/PD Target (*n* = 37)	Patients on PK/PD Target (*n* = 51)	Patients over the PK/PD Target (*n* = 53)	*p*-Value
Sex (male/female)	24/13 (65/35)	35/16 (69/31)	31/22 (58/42)	0.55
Age (years)	65 (52–75)	71 (56–78)	75 (65–81)	0.002
Weight (kg)	75 (65.5–84.75)	78 (65–87)	70 (64.25–85)	0.50
Height (cm)	173 (163–179)	172 (165–179)	170 (163–175)	0.24
BMI (kg/m^2^)	25.3 (23.1–27.7)	25.7 (24.1–28.9)	24.4 (23.2–27.7)	0.54
Serum total protein (g/L)	30.0 (26.4–33.5)	29.0 (25.0–34.0)	27.0 (23.0–30.7)	0.09
Serum albumin (g/L)	55.0 (52.0–63.0)	56.5 (52.5–65.2)	54.0 (49.0–58.0)	0.10
Alanine aminotransferase (UI)	21.0 (10.0–48.0)	25 (15.0–43.0)	28.0 (14.2–41.0)	0.68
Serum creatinine (mg/dL)	0.74 (1.01–0.58)	0.76 (0.56–1.18)	1.3 (0.95–1.97)	<0.001
eCLCr ^1^ (mL/min)	104.9 (68.9–136.5)	96.6 (47.3–139.2)	53.1 (33.8–79.5)	<0.001
C-reactive protein (mg/L)	27.1 (7.7–89.1)	59.3 (21.5–111.8)	75.1 (32.9–164.8)	0.005
Time between initiation therapy with meropenem and first TDM (hours)	72 (72–120)	96 (72–96)	96 (72–120)	0.80
Meropenem daily dose (g)	3 (2–3)	3 (2–4)	3 (2–4)	0.39
Serum meropenem concentration/daily dose ratio (C/D)	2.3 (1.0–2.6)	3.7 (2.6–6.7)	8.5 (5.9–10.8)	<0.001
Serum meropenem concentration/kg (C/kg)	0.08 (0.01–0.10)	0.16 (0.12–0.20)	0.33 (0.23–0.4)	<0.001
Serum meropenem concentration normalized for daily dose and weight (C/D/kg)	0.03 (0.01–0.04)	0.05 (0.03–0.09)	0.11 (0.08–0.14)	<0.001

^1^ Data are presented as count (percentage) or median (interquartile range).

## Data Availability

The data presented in this study are available on request from the corresponding author.

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
