# Peer review of "Meropenem PK/PD Variability and Renal Function: “We Go Together”"

_pharmaceutics, 2023, doi:10.3390/pharmaceutics15092238_

Round 1

Reviewer 1 Report

please provide higher resolution figures

Author Response

Dear Reviwer 1,

Thank you for your comment. I replaced the previous figure with a better resolution new figure .

Best regards,

J

Reviewer 2 Report

Meropenem, a widely used antibiotic, is impacted by renal function variability among patients, necessitating therapeutic drug monitoring (TDM) for personalized dosing. TDM-guided treatment is vital in combating severe infections and antimicrobial resistance, spanning various clinical settings to enhance efficacy in antibiotic administration. The authors described it in details and the writing is good. It can be accepted after the following queries being addressed.

Please clarify the following:

1. What will be the effect of similar alternative drugs for meropenem?

2. What will be the impact of the study in different geographical locations? 

3. It would be better if the authors could include some experimental data e.g., HPLC, etc. to support the claims. Otherwise, there will be always doubt in the reader's mind about the conclusions drawn in this type of study. (The authors also mentioned the same) 

4. Please fix Figure 1, the resolution is very poor.

Reviewer 3 Report

This study on meropenem is recommended for publication in Pharmaceutics for its clinical meanings. Some  comments are shown as followings:

1. How did the authors calculate the correlation between creatinine clearance and meropenem exposure?

2. Figure 1b is blurred.

3. Statistical difference analysis should be provided for Figure 3 and 4.
